# Structure of the Mating-Type Genes and Mating Systems of *Verpa bohemica* and *Verpa conica* (Ascomycota, Pezizomycotina)

**DOI:** 10.3390/jof9121202

**Published:** 2023-12-15

**Authors:** Wenhua Sun, Wei Liu, Yingli Cai, Xiaofei Shi, Liyuan Wu, Jin Zhang, Lingfang Er, Qiuchen Huang, Qi Yin, Zhiqiang Zhao, Peixin He, Fuqiang Yu

**Affiliations:** 1School of Food and Biological Engineering, Zhengzhou University of Light Industry, Zhengzhou 450002, China; vanwenhua@163.com (W.S.); yinqihtw@163.com (Q.Y.); 2Germplasm Bank of Wild Species, Yunnan Key Laboratory for Fungal Diversity and Green Development, Kunming Institute of Botany, Chinese Academy of Sciences, Kunming 650201, China; zhenpingliuwei@mail.kib.ac.cn (W.L.); shixiaofei@mail.kib.ac.cn (X.S.); wuliyuan0012@163.com (L.W.); zhangjinjinshang02@163.com (J.Z.); elingfang@163.com (L.E.); huangqiuchen0816@yeah.net (Q.H.); 3Institute of Agro-Products Processing, Yunnan Academy of Agricultural Sciences, Kunming 650221, China; loveylcai@163.com; 4College of Resource and Environment, Yunnan Agricultural University, Kunming 650100, China; 5School of Biodiversity Conservation, Southwest Forestry University, Kunming 650224, China; 6Agricultural Technology Promotion Station in Zhuoni County, Gannan 747600, China; znzhiqiang@163.com

**Keywords:** fungi, life cycle, homothallism, heterothallism, *MAT1-1-1*, *MAT1-2-1*, *MAT1-1-10*, *MAT1-1-11*

## Abstract

*Verpa* spp. are potentially important economic fungi within Morchellaceae. However, fundamental research on their mating systems, the key aspects of their life cycle, remains scarce. Fungal sexual reproduction is chiefly governed by mating-type genes, where the configuration of these genes plays a pivotal role in facilitating the reproductive process. For this study, de novo assembly methodologies based on genomic data from *Verpa* spp. were employed to extract precise information on the mating-type genes, which were then precisely identified in silico and by amplifying their single-ascospore populations using MAT-specific primers. The results suggest that the *MAT* loci of the three tested strains of *V. bohemica* encompassed both the *MAT1-1-1* and *MAT1-2-1* genes, implying homothallism. On the other hand, amongst the three *V. conica* isolates, only the *MAT1-1-1* or *MAT1-2-1* genes were present in their *MAT* loci, suggesting that *V. conica* is heterothallic. Moreover, bioinformatic analysis reveals that the three tested *V. bohemica* strains and one *V. conica* No. 21110 strain include a *MAT1-1-10* gene in their *MAT* loci, while the other two *V. conica* strains contained *MAT1-1-11*, exhibiting high amino acid identities with those from corresponding *Morchella* species. In addition, MEME analysis shows that a total of 17 conserved protein motifs are present among the *MAT1-1-10* encoded protein, while the *MAT1-1-11* protein contained 10. Finally, the mating type genes were successfully amplified in corresponding single-ascospore populations of *V. bohemica* and *V. conica*, further confirming their life-cycle type. This is the first report on the mating-type genes and mating systems of *Verpa* spp., and the presented results are expected to benefit further exploitation of these potentially important economic fungi.

## 1. Introduction

*Verpa* is an epigeous genus within Morchellaceae (Ascomycota, Pezizomycotina). The phylogenetic analysis of ribosomal DNA revealed a close relationship between it and the genus *Morchella* [1,2]. Hence, *Verpa* spp. are also called false or early morels, due to similar morphology of their ascocarp and earlier occurrence in spring in comparison with true morels (*Morchella* spp.) [3]. *Verpa* spp. are widespread in the northern temperate zones and were originally found in the northern parts of North America, Europe, and Asia. Globally, although they are not as common as true morels, the false morels are often found in a wide range of habitats and environmental conditions [4], even in the wild mushroom market. A total of 28 *Verpa* species are recorded in Index Fungorum (http://www.indexfungorum.org/Names/Names.asp, accessed 1 November 2023), among which, two species—*V. bohemica* and *V. conica*—have been extensively studied.

*V. bohemica* is locally used for dietary and antioxidant purposes in tribal areas of Kashmir Himalaya. However, when eaten in excess, they may cause gastrointestinal upset and loss of muscular coordination [5]. Thus, *V. bohemica* was eventually recognized as an inedible mushroom [6]. Meanwhile, many studies have indicated that *V. bohemica* presents quite impressive antibacterial and antifungal activities [5], as well as strong antioxidant activities [7,8]. Furthermore, high contents of lovastatin, γ-aminobutyric acid (GABA), and ergothioneine have been found in mycelial and single-cell biomasses of *V. bohemica*, implying that the fungus possesses effective anti-inflammatory, antioxidant, profibrinolytic, and antihypertensive effects [9]. On the other hand, apart from its strong antioxidant activity in different systems [10], *V. conica* has exhibited strong laccase-like multicopper oxidase activity enhancement in the presence of the naturally occurring phenolic compound guaiacol, showing great potential for plant litter decay [11]. However, there is scarce research on the genetic information, life cycle, and reproductive system of *Verpa* spp. in the existing literature.

The genetic locus responsible for the determination of the mating type in fungi is referred to as the mating-type locus. Ascomycetes regulate their sexual reproduction through a single mating-type locus [12] which contains idiomorphs [13], distinct and non-homologous mating-type “alleles” in heterothallic species which can encompass one or multiple genes, specifically known as *MAT1-1* and *MAT1-2*, respectively [14]. The *MAT1-1* idiomorph harbors the *MAT1-1-1* gene, which encodes a protein containing an MATalpha_HMGbox domain. Similarly, the *MAT1-2* idiomorph harbors the *MAT1-2-1* gene, which encodes a protein containing a high-mobility group (HMG) domain [15]. In heterothallic ascomycete species, a single ascospore usually contains only one *MAT* idiomorph, *MAT1-1* or *MAT1-2*, while in homothallic ascomycetes of the same lineage, there will be two mating types in the same genome, but the arrangement will often differ [15,16]. Apart from the core mating-type genes (*MAT1-1-1* and *MAT1-2-1*), other mating-type genes on the *MAT* locus are lineage-specific and conserved at the family and class levels. To date, a total of 12 *MAT1-1* genes (*MAT1-1-1* to *MAT1-1-12*) and 15 *MAT1-2* genes (*MAT1-2-1* to *MAT1-2-15*) have been identified across various species [17,18].

While research on the mating type gene of *Verpa* has been limited, studies on this gene in closely related species such as *Morchella* have been more extensive. Liu et al. [19] have employed genome data to elucidate the heterothallic life cycle of *Morchella importuna*. Chai et al. [20] identified two new secondary *MAT* genes in *Morchella* spp.; namely *MAT1-1-10* and *MAT1-1-11*. Expanding on their findings, Chai et al. [21] examined the structure of the mating-type loci across ten *Morchella* species, revealing that eight were heterothallic, while two were homothallic.

In this study, the whole genome of the two *Verpa* species (three strains per species) were sequenced, assembled, and annotated with the purpose of: (1) identifying the *MAT* loci of the two *Verpa* species, (2) analyzing the differences in *MAT* locus structure between *Verpa* and *Morchella*, and (3) determining the reproductive strategy of *Verpa* species (i.e., homothallic or heterothallic).

## 2. Materials and Methods

### 2.1. Fungal Cultivation and Ascosporic Isolates

Six strains of *V. bohemica* and *V. conica* were sourced from five provinces in China (Table 1). To obtain pure mycelium culture, tissue isolation was performed from fresh or naturally dried specimens. The specific procedure was as follows: A tissue block of approximately 0.5 cm^2^ in size was taken from the stipe part of the ascomata. After washing with sterile water 2–3 times, the block was disinfected with 75% alcohol for 1.5 min. Then, the tissue was sliced into 1–2 mm^3^ pieces and placed on suitable culture media (PDA, 200 g of potato extract, 20 g of glucose, and 20 g of agar per liter of medium) for constant temperature cultivation at 21 °C. The emerging tips of individual tissue blocks were then excised and transferred to a new PDA plate. Three successive cultivation cycles later, strains of a purified nature were successfully derived.

Among six *Verpa* specimens, the fruiting bodies of No. 20020 (*V. bohemica*) and No. 21120 (*V. conica*), which produced a large number of mature ascospores, were selected for the isolation of monosporic cultures. After the process of gradient dilution [22], three consecutive low-concentration gradient spore suspensions were independently applied onto three separate PDA plates and subjected to the same incubation conditions mentioned earlier. The emerging tips of individual ascospores were then excised and transferred to a new PDA plate under a stereomicroscope (Leica M205, Wilmington, DE, USA). Each monosporic isolate was assigned a corresponding Arabic numeral suffix. For example, 20020-1 refers to the first ascosporic isolate. All cultures were stored in the Germplasm Repository of the Kunming Institute of Botany, Chinese Academy of Sciences, Yunnan Province, China.

### 2.2. DNA Extraction and Sequencing Scheme

Fresh mycelium grown on PDA plates for 7 days were collected for DNA and RNA extraction. DNA was extracted from fungal mycelium using the modified cetyltrimethylammonium bromide (CTAB) method [23], and RNA contamination was eliminated using RNAse A digestion. Total RNA was extracted using TRIzol^®^ Reagent (Invitrogen, Carlsbad, CA, USA) and purified with Plant RNA Purification Reagent (Invitrogen, Carlsbad, CA, USA). Both the extracted DNA and RNA were subjected to subsequent sequencing analysis.

A hybrid scheme of Illumina sequencing and Nanopore sequencing was utilized. For Illumina sequencing on the Illumina HiSeq4000 platform (San Diego, CA, USA), two paired-end (PE) libraries with insert fragment sizes of 270 bp and 500 bp, respectively, were constructed, and Illumina PE150 sequencing was performed, generating approximately 4 gigabases of raw data for each library. In the Nanopore sequencing approach, each sample was sequenced using the sensitive flow cells of the PromethION sequencer and underwent sequencing reactions to produce at least 5 gigabases of raw data. Additionally, RNA sequencing was conducted on the mycelia of each strain using the Illumina HiSeq4000 (San Diego, CA, USA) sequencing platform, with a PE library having an insert size of 270 bp and PE150 paired-end sequencing, generating 4 gigabases raw data for subsequent gene structure annotation. The original sequencing data were submitted to the Sequence Read Archive (SRA) database with access numbers SRR22519296–SRR22519313.

### 2.3. Genome Assembly and Annotation

The raw reads generated by Illumina sequencing were subjected to quality control using Trimmomatic v0.39 [24], filtering out low-quality reads, adapters, and reads containing ‘N’. Nanopore reads were trimmed using nanofilt 2.8.0 [25]. Entire genome assembly was completed using a hybrid approach that combined Illumina sequencing and Nanopore clean sequencing data. The de novo genome assembler suite, comprising NextDenovo v2.4.0 and NextPolish v1.3.1 (both developed by NextOmics; https://github.com/Nextomics, accessed on 1 December 2023), was utilized to produce highly accurate and continuous genome assembly. Gene prediction was performed using geta v2.6.1 (https://github.com/chenlianfu/geta, accessed on 1 December 2023), combined with ab initio, homologous protein alignment and RNA sequencing transcription alignment, followed by Nr annotation using a local Nr database, and custom scripts were utilized to organize and summarize the results into the final gene annotation file. The final de novo genomic assembly results were submitted to the National Center for Biotechnology Information (NCBI), and their respective public accession numbers can be found in Table 1.

### 2.4. Determination of the MAT Locus Structures

In the six available *Verpa* genome sequences, each *MAT* gene was selected based on the feature descriptions in the genome annotation files. Each *MAT* protein and its neighboring region were further confirmed through BLASTp with published homologous proteins. The functional domains of *MAT* genes were determined using a conservative domain search available on NCBI. Additionally, the domain alignment was depicted using the R package ggmsa (http://yulab-smu.top/ggmsa/, accessed on 1 December 2023).

### 2.5. Comparison of MAT Loci

Comparison was conducted on the *MAT* gene loci and their adjacent regions mined from the genome sequences of *Verpa* spp., and visualization was performed using R version 4.0.2 and the gggenes package version 0.4.1 (https://github.com/wilkox/gggenes, accessed on 1 December 2023). To create linear gene maps, protein gene coordinates were incorporated into a .csv file and imported into the gggenes package in RStudio [26]. The resulting genome maps of *MAT* gene loci were exported as .pdf files. The sequences of the two *Morchella* species [19] used for comparison were obtained from public genomic databases at NCBI with accession numbers QORM00000000 and QOKS00000000, respectively.

### 2.6. Motif Identification of Secondary MAT Genes

Conserved amino acid sequences of secondary *MAT* proteins were predicted using the online MEME tool (Multiple Em for Motif Elicitation) [27]. The XML file obtained from the online MEME tool was obtained by visualizing the Batch MEME Motifs Viz plugin in TBtools (v1.123) [28].

### 2.7. Phylogenetic Analysis

To confirm the accuracy of the *MAT* proteins in *Verpa*, published *MAT* protein sequences were selected to construct a phylogenetic tree [17,18]. All sequences underwent alignment using MAFFT version 7.508 [29] with options --maxiterate 1000 --retree 1 --localpair. Maximum likelihood (ML) phylogenies were generated from the alignment in IQ-TREE v2.2.0 [30]. Branch supports were assessed using the ultra-fast bootstrapping method with 1000 replicates [31].

### 2.8. Primer Design for MAT Genes

Primers of *MAT1-1-1* and *MAT1-2-1* were designed to determine the sexual reproduction mode of *Verpa* spp., additionally, primers were also designed to amplify *MAT1-1-10* and *MAT1-1-11*. All primers were designed using Primer Premier v.5 with the optimal sequence length of 20 bp, optimal annealing temperature of 60 °C, and the fragment length prioritizing the crossing of a certain intron. Primers were synthesized by Beijing Tsingke Biotechnology. PCR amplification was performed in a 15 μL reaction mixture, including 7.5 μL of PCR mix (Beijing Tsingke Biotechnology, Beijing, China), 0.375 μL of each primer (10 μM), and 1 μL of genomic DNA (25 ng/μL). The PCR products were analyzed using 1.5% agarose gel electrophoresis. The primer sequences and key PCR cycling parameters are listed in Table 2.

## 3. Results

### 3.1. Structure of the MAT1–1-1 and MAT1–2-1 Genes of V. bohemica and V. conica

The gene structure and length vary inter-species and intra-species of *Verpa*. The predicted *MAT1-1-1* genes in *V. bohemica* strains No. 20020, No. 20124, and No. 21108, as well as in *V. conica* strain No. 21110, each exhibited the presence of two introns, with the intron length as 57–66 and 56–57 bp, respectively. The lengths of these genes were different, measuring 1112 bp, 1112 bp, 1163 bp, and 1182 bp, respectively. The predicted *MAT1-1-1* genes in *Verpa* spp. encoded proteins with the following amino acid sequences: 329 aa (No. 20020), 329 aa (No. 20124), 346 aa (No. 21108), and 355 aa (No. 21110) (Appendix A). The inferred amino acid sequences of these four *Verpa* strains all possessed a conserved MATalpha_HMGbox domain (GenBank: cl07856) with E-values ranging from 9.15 × 10^−5^ to 2.7 × 10^−5^ (Figure 1a). In the case of *V. bohemica*, all three predicted MATalpha_HMGbox domain spanned the first intron with the length of 66 bp, whereas it was interrupted by a 57 bp intron in *V. conica* No. 21110. The MATalpha_HMGbox domain in *Verpa* spp. contained an intron that spanned a conserved arginine codon, as indicated by the red arrow in Figure 1a.

Similarly, the predicted *MAT1-2-1* genes in *V. bohemica* strains No. 20020, No. 20124, and No. 21108, as well as in *V. conica* strains No. 21117 and No. 21120, each exhibited the presence of three introns. These genes exhibited varying lengths—specifically, 1073 bp, 1073 bp, 1073 bp, 1081 bp, and 1081 bp—and their coding regions translated into protein sequences of 281 aa (No. 20020), 281 aa (No. 20124), 281 aa (No. 21108), 305 aa (No. 21117), and 305 aa (No. 21120), respectively (Appendix A). Additionally, alignment analysis of the aforementioned sequences revealed that a conserved HMG-box domain (GenBank: cd01389) with E-value from 2.8 × 10^−21^ to 1.6 × 10^−21^ in the *MAT1-2-1* protein of *Verpa* spp. was identified (Figure 1b). All of the predicted *MAT1-2-1* genes of *V. bohemica* had an HMG-box domain interrupted by two introns, whereas *V. conica* featured an HMG-box domain spanning three introns. The HMG-box domain of in *Verpa* spp. contained an intron that spanned a conserved serine codon, as indicated by the red arrow in Figure 1b.

The amino acid sequence also varies inter-species and intra-species of *Verpa*. The amino acid sequences in the C-termini regions exhibited high variability (Appendix A). In stark contrast, the *MAT1-2-1* protein was conserved among different species, even in the C-termini regions, with variations occurring only in specific regions, such as 119 aa to 142 aa in *V. conica* No. 21117 and 128 aa to 151 aa in No. 21120 (Appendix A).

### 3.2. Structure of the MAT1–1-10 and MAT1-1-11 Genes of V. bohemica and V. conica

Within the *MAT* loci of the six investigated strains, aside from *MAT1-1-1* and *MAT1-2-1*, additional open reading frames (ORFs) were detected. For *V. bohemica*, the top BLASTp hits for Gene05695 of No. 20020, Gene05694 of No. 20124, and Gene05692 of No. 21108 were all the *MAT1-1-10* protein of *M. sextelata* (QQL94615) with e-values of 1 × 10^−130^, 7 × 10^−131^, and 9 × 10^−109^, respectively (Appendix A). The *MAT* loci of *V. conica* strains No. 21117 and No. 21120 exclusively contained the *MAT1-2* idiomorph, lacking the *MAT1-1* idiomorph, and both harbored another ORF. In the top BLASTp results for Gene05376 of No. 21117 and Gene05383 of No. 21120, apart from the two highest-ranking “hypothetical proteins” in the NCBI database, the best BLASTp hit corresponded to the *MAT1-1-11* protein of *M. sextelata* (QQL94614) (Appendix A). Following the nomenclature guidelines outlined by Turgeon et al. [14] and Wilken et al. [17], we designated Gene05695 of No. 20020, Gene05694 of No. 20124, Gene05692 of No. 21108, and Gene05312 of No. 21110 as *MAT1-1-10*, and Gene05376 of No. 21117 and Gene05383 of No. 21120 as *MAT1-1-11*.

All three strains of *V. bohemica* possessed *MAT1-1-10* genes with five introns, each with a length of 1782 bp. However, the *MAT1-1-10* gene of *V. conica* strain No. 21110 contained three introns with a combined length of 1686 bp. The *MAT1-1-10* genes in *Verpa* spp. encoded proteins with the amino acid lengths of 498 aa (No. 20020), 498 aa (No. 20124), 441 aa (No. 21108), and 507 aa (No. 21110). These hypothetical *MAT1-1-10* proteins from three *V. bohemica* strains showcased amino acid identities surpassing 35% with the *Morchella* genus, while those of *V. conica* strain No. 21110 had amino acid identities of 38.60% (e-value = 1 × 10^−114^) and 37.94% (e-value = 2 × 10^−104^) with *Morchella* sp. *Mes-20* (AVI61143) and *M. rufobrunnea* (QQL94651), respectively (Appendix A). In contrast, the *MAT1-1-11* genes found in *V. conica* possessed three introns measuring 1619 bp and 1995 bp in length, with the coding region encoding proteins of 445 and 424 amino acids, respectively (Appendix A). *MAT* loci of *V. conica* strains No. 21117 and No. 21120 solely encompassed the *MAT1-2* idiomorph, lacking the *MAT1-1* idiomorph, and both harbored a hypothetical *MAT1-1-11*. BLASTp analysis revealed that Gene05376 of No. 21117 and Gene05383 of No. 21120 exhibited amino acid identities of 45.59% (e-value = 4 × 10^−114^) and 45.59% (e-value = 1 × 10^−113^) with *MAT1-1-11* from the *Morchella* genus (Appendix A).

### 3.3. Phylogenetic Analysis of MAT Genes

As shown in Figure 2, identical *MAT* proteins were clustered together. The analysis revealed that the *MAT1-1-1* genes from the genus *Verpa* formed a cluster with 100% support, subsequently clustering with those of the genus *Morchella*, both belonging to Morchellaceae, and collectively with the genus *Tuber* (Tuberaceae) and *Phymatotrichum* (Rhizinaceae) in the order Pezizales. The corresponding *MAT1-1-10* and *MAT1-1-11* genes of the genus *Verpa* also clustered with those of the genus *Morchella*, with both genes clustered under independent branches under 98% and 100% support, respectively (Figure 2a). Moreover, the *MAT1-2-1* gene from *Verpa* exhibited a high-support cluster with its counterpart in *Morchella, Tuber* and *Phymatotrichum* (Figure 2b). This evidence substantiates the accuracy of the designated nomenclature for *MAT* genes within *Verpa* spp.

### 3.4. Analysis of Conserved Motifs of MAT1-1-10 and MAT1-1-11 Proteins

Incorporating the revelations derived from *Morchella* genus by Chai et al. [21], motif analysis was employed to characterize the domains within the *MAT1-1-10* and *MAT1-1-11* proteins. Through MEME analysis, a total of 17 recognized protein motifs were unveiled within the *MAT1-1-10* protein, while the *MAT1-1-11* protein contained 10 conserved protein motifs. Subsequently, the motifs were arranged in ascending order according to their e-values, as exemplified in Figure 3.

Within the 17 conserved protein motifs discovered in the *MAT1-1-10* protein, motifs 14, 15 and 16 exhibited specificity towards *Verpa* spp., while motifs 13 and 17 demonstrated specificity towards *Morchella* spp.; furthermore, among *Verpa* spp., motifs 3 and 4 were exclusively present in *V. bohemica* and absent in *V. conica*. Similarly, variations were noted among diverse *Morchella* species in relation to the *MAT1-1-10* protein. Specifically, motifs 13 and 17 were present in Esculenta clade species (*Morchella* sp. *Mes-20*, *Morchella* sp. *Mes-6*, *Morchella* sp. *Mes-19*, and *M. crassipes*), whereas they were absent in Elata clade species, including *M. sextelata*, *M. importuna*, and *M. purpurascens*. To establish the prevalence of this phenomenon among *Morchella* species, further substantiating data are required. Overall, the *MAT1-1-10* proteins exhibited remarkably similar structures within the confines of the same species, although certain species lacked specific regions (as in *M. crassipes*). Conversely, significant conservation was observed at the 5′ and 3′ ends of the sequences among different species, while the region proximal to the 3′ end exhibited variability. Hence, if we define a domain for the *MAT1-1-10* protein, we consider it reasonable to include motif 6, motif 1, motif 11, motif 2, motif 9, and motif 8 within this region. In fact, this is a continuous sequence that exists at the 5′ end of *MAT1-1-10*, where motif 1 and motif 2 have the highest e-values (Figure 3a). With regard to the *MAT1-1-11* protein, out of the 10 conserved protein motifs, motif 7 and motif 10 were exclusively identified in *Morchella* species. Variations were detected at the 3′ end of the sequence, while other regions remained relatively conserved. Consequently, it is justifiable to designate the middle region as the domains of the *MAT1-1-11* protein, such as the region composed of motif 2, motif 9, and motif 1, based on their coexistence and conservatism (Figure 3b). However, we maintain the perspective that, in the delineation of a domain for the *MAT1-1-10* and *MAT1-1-11* proteins, motif 1 should both be deemed as their central regions.

### 3.5. MAT Locus Gene Content of Verpa

With regard to the genus *Morchella*, which exhibited the closest phylogenetic relationship to the genus *Verpa*, we constructed a gene structure map of the *MAT* locus of *Verpa* spp., as compared with the published genome data of morels [19] and found that its gene structure differed. Here, we define the direction of the universal *APN2* to be upstream of the *MAT* locus, while the *SLA2* gene is downstream. In these cases, *MAT* gene order in *Verpa* and *Morchella* maintained consistency, but their quantity and location clearly differed. *MAT1-1-1* and *MAT1-2-1* had the same gene transcription direction, deviating from *MAT1-1-10* and *MAT1-1-11* in these fungi. The *MAT* loci of the three *V. bohemica* simultaneously contained both *MAT1-1-1* and *MAT1-2-1*, and each possessed a *MAT1-1-10* gene (Figure 4), indicating that it is a homothallic fungus at the molecular level. However, *V. conica* strain No. 21110 showed similarity to *V. bohemica* in *MAT* loci genes, but lacked the *MAT1-2-1* gene, only containing *MAT1-1-10* and *MAT1-1-1* genes. On the other hand, strains No. 21117 and No. 21120 only contained *MAT1-2-1* and also included an additional gene, *MAT1-1-11* (Figure 4), suggesting that it has a heterothallic life cycle. In contrast, within the *Verpa* genus, *APN2* and *SLA2* genes are closely adjacent to the *MAT* locus. Between *APN2* and *SLA2*, no additional coding genes were inserted except for the identified mating-type genes in *Verpa* spp. *MAT* locus. The actual situation could be inferred as compared to *Morchella*: the sequences at both ends (*SDH2* to *APN2*, *SLA2* to *MBA1*) of the *Verpa MAT* loci undergo complete in situ flipping (Figure 4). Furthermore, we adhered to the naming recommendations for the upstream and downstream genes of the *MAT* locus in *M. importuna* proposed by Du and Yang [32] and made uniform modifications, such as changing *atp-3/ATPs4* to *ATP4* and *Mba1/Mab1* to *MBA1*. We also discovered proteins homologous to *ATP4* and *MBA1* in the genus *Verpa*. Nonetheless, the molecular functions of the genes flanking the *MAT* loci remain unknown.

### 3.6. PCR Amplifications of MAT Genes

Primer pairs of the *MAT1-1-1* (b1) and *MAT1-2-1* (b2) genes were used to detect the mating type of single-ascospore populations in *V. bohemica*. Similarly, primer pairs of *MAT1-1-1* (c1) and *MAT1-2-1* (c2) genes were used for those in *V. conica*. Additionally, special primers were designed to amplify the *MAT1-1-10* and *MAT1-1-11* genes in *V. bohemica* and *V. conica* (Table 2). Among the 47 single-ascospore populations of *V. bohemica*, both the *MAT1-1-1* and *MAT1-2-1* gene were successfully amplified in each ascospore strain (Figure 5a,b). In the 47 single-ascospore populations of *V. conica*, strains that contained the *MAT1-1-1* gene but lacked the *MAT1-2-1* gene were amplified and identified as *MAT1-1-1* gene, and vice versa. Two genes exhibited perfect complementarity within the single-ascospore population (Figure 5c,d). Additionally, primers for b10, c10, and c11 effectively amplified the *MAT1-1-10* gene of *V. bohemica*, the *MAT1-1-10* gene of *V. conica*, and the *MAT1-1-11* gene of *V. conica*, respectively (Appendix A). Furthermore, we attempted to verify the presence of the *MAT1-1-11* gene in *V. bohemica* using the primer pair c11, designed for the *MAT1-1-11* gene of *V. conica*. The results revealed that the *MAT1-1-11* gene is not present in *V. bohemica*, consistent with the genomic data. In this study of *V. bohemica* and *V. conica*, among the five mating-type genes, no sequence differences were detected between the single-ascospore strains and the parental strains.

## 4. Discussion

### 4.1. MAT Loci and Mating-Type Genes

Through our investigation, it was determined that all three strains of *V. bohemica* consistently exhibited a configuration of the *MAT* locus which includes three ORFs: the *MAT1-1-1*, *MAT1-1-10*, and the *MAT1-2-1* genes. In contrast, *V. conica* exhibited an entirely distinct pattern, with each strain exclusively harboring one type of *MAT* idiomorph—either *MAT1-1* or *MAT1-2*. Strain No. 21110 exclusively possessed the *MAT1-1* idiomorph, encompassing the *MAT1-1-1* and *MAT1-1-10* genes. Conversely, strains No. 21117 and No. 21120 exclusively contained the *MAT1-2* idiomorph, housing the *MAT1-2-1* and *MAT1-1-11* genes. This is the first report of the *MAT1-1-10* and *MAT1-1-11* genes in species other than those in the genus *Morchella* since Chai et al. [20] initially identified them in *Morchella*.

Compared to other protein-coding genes, sex-determining genes in ascomycetes have been found to evolve at a faster rate, making them useful for studying the phylogenetic relationships of fungi [33]. Previous studies on the phylogenetic analysis of mating-type genes have been conducted in ascomycetes [34,35]. In this study, phylogenetic trees were constructed using different *MAT1-1* and *MAT1-2* protein sequences, in order to corroborate the accuracy of the naming of *MAT* genes in the genus *Verpa*.

Glass et al. [36] first described the *MAT1-1-1* gene in the matA (=*MAT1-1*) idiomorph of *N. crassa*, which encodes a DNA-binding protein with a conserved MATalpha_HMGbox domain. While previous research has suggested that the *MAT1-1-1* gene typically contains an intron within its conserved domain [37,38,39], subsequent studies have revealed exceptions to this pattern (as in *Sporothrix schenckii* [40]). Nevertheless, in this study, the *MAT1-1-1* gene in *Verpa* spp. clearly supported the previously established viewpoint (Figure 1a). The *MAT1-2-1* gene harbored a conserved HMG-box domain, which has been demonstrated to play a role in DNA binding across plants, animals, and fungi [41,42,43]. In accordance with prior research, the ORF of the *MAT1-2-1* gene in *Verpa* spp. included an intron spanning a serine codon (Figure 1b). The definition of a dependable domain for the *MAT* gene has been a persistent challenge. The *MAT1-1-2* gene—the second *MAT1-1* gene described—was exclusively present in Sordariomycetes [17]. Previous studies have aimed to establish a reliable structural domain for the *MAT1-1-2* protein [15,37], which now has been included in the Protein Family (PFAM) database. Earlier studies suggested a role for the *MAT1-1-2* gene in the development of sexual structures [44,45,46]; however, its precise function in sexual reproduction has remained enigmatic. This study attempted to define a motif for the recently discovered secondary *MAT* genes, *MAT1-1-10* and *MAT1-1-11*, with the aim of contributing a novel domain definition for future research. However, it is crucial to note that this motif relies solely on data available in *Morchella* and *Verpa*. Subsequent discoveries are imperative to validate the applicability of this domain and confirm the functionality of this motif, necessitating further investigation.

### 4.2. Structure of Mating-Type Loci between Verpa bohemica and Verpa conica

The gene structure of the *MAT* idiomorph in *Verpa* spp. exhibited a resemblance to the corresponding structure in *Morchella*. For instance, the *MAT* idiomorph of *V. bohemica* No. 21110 and M04M24 of *M. importuna*, as reported by Liu et al. [19], shared a similar configuration. Similarly, the *MAT* idiomorph of *V. bohemica* aligned with the structure of *M. rufobrunnea*, as published by Chai et al. [21], where the *MAT1-1-10*, *MAT1-1-1*, and *MAT1-2-1* genes were arranged in a tandem array. However, discernible structural disparities existed between the *MAT* loci of these two species. For example, *M. importuna* also possesses an alternative structure, consisting solely of one *MAT1-2-1* gene in the *MAT1-2* idiomorph [19]. Recent research has suggested that the *MAT1-2* idiomorph of *M. purpurascena* and *M. pulchella* encompasses the *MAT1-2-1* and *MAT1-1-11* genes [21], corresponding to the structure observed in strains No. 21117 and No. 21120 of *V. conica*, as delineated in this investigation. However, the presence of a *MAT1-1* gene in the *MAT1-2* idiomorph remains a perplexing phenomenon. Similarly, the *MAT1-2* idiomorph of *Chaetomium globosum* (Sordariomycetes) included a *MAT1-1-2* gene located 1540 bp downstream of the *MAT1-2-1* gene [15].

Butler et al. [47] were the first to propose the concept of *MAT* context in ascomycetes through their investigation of various yeasts and *N. crassa*. They observed that the *MAT* locus in *Yarrowia lipolytica* was located between the homologous genes *APN2* and *SLA2* in *S. cerevisiae*, and these two genes were also identified in the left and right adjacent sequences in *N. crassa*. Consequently, this shared structural arrangement led the authors to suggest that this configuration may be ancestral to all ascomycetes. Debuchy et al. [15] supported this conclusion and additionally proposed that *APN2*, *SLA2*, and *APC5* are ancestral partners of the *MAT* locus in ascomycetes. However, this study raised questions about whether *APC5* truly fits this description. In summary, a more comprehensive examination of the regions surrounding the *MAT* locus is expected to contribute to better understanding of the ancestral *MAT* locus structure in ascomycetes.

### 4.3. Homothallism in V. bohemica and Heterothallism in V. conica

In contrast to animals with sex chromosomes, the mechanism of sexual reproduction in fungi is governed by a solitary genetic locus known as the mating-type locus [48,49]. Compared to basidiomycetes, the mating-type loci of ascomycetes only contain a few relatively conserved genes, usually *MAT1-1* and/or *MAT1-2* [50]. Within the genomic data of the three strains of *V. bohemica*, it was ascertained that this species harbors both the *MAT1-1-1* and *MAT1-2-1* gene in the *MAT* locus, signifying the homothallic characteristic of *V. bohemica*. Concerning *V. conica*, genomic data of the three strains revealed that No. 21110 exclusively contained the *MAT1-1* idiomorph, while No. 21117 and No. 21120 solely contained the *MAT1-2* idiomorph, manifesting characteristics akin to heterothallic ascomycetes. In fact, the missing genotypes in the *V. conica MAT* locus could not be detected throughout the entire genome, either in silico or by PCR amplification. The conserved flanking sequences of these two distinct types validated their occupancy of the same chromosomal locus. PCR analysis further substantiated these findings, as we effectively amplified the *MAT1-1-1*, *MAT1-2-1*, and *MAT1-1-10* genes within the single-spore strain of *V. bohemica* No. 20020. In contrast, within the single-spore population of *V. conica* No. 21120, *MAT1-1-1* and *MAT1-2-1* demonstrated complementary target products, while *MAT1-1-10* and *MAT1-1-11*, which are intrinsically associated with these two genes, exhibited an identical banding pattern corresponding exclusively to their counterparts. Therefore, these results strongly indicate the homothallic life cycle of *V. bohemica*, while *V. conica* exhibits heterothallism. Discovery of the life cycle of *Verpa* spp. through genomic data analysis holds promise for providing further insights into their future utilization.

## 5. Conclusions

The results of this study revealed heterothallism in *V. conica* and homothallism in *V. bohemica*, indicative of the existence of both reproductive strategies within the *Verpa* genus. Leveraging genomic data enabled a comprehensive analysis of mating-type loci across six *Verpa* species. It was found that *V. bohemica*, a homothallic species, includes both *MAT1-1-1* and *MAT1-2-1* genes alongside an independent *MAT1-1-10* within its mating-type loci, while *V. conica*, a heterothallic species, possesses either *MAT1-1-1* or *MAT1-2-1*, with *MAT1-1-10* and *MAT1-1-11* genes specific to the *MAT1-1* and *MAT1-2* idiomorphs, respectively. Consistent with the pattern observed in other ascomycetes, the *MAT1-1-1* gene in *Verpa* spp. demonstrated greater variability than the *MAT1-2-1* gene, particularly within their conserved domains. Contrary to *Morchella*, where the *SDH2* and *MBA1* genes are adjacent to the *MAT* locus, the *Verpa* spp. displayed a significantly different flanking gene composition, with *APN2* and *SLA2* in proximity and *SDH2* and *MBA1* positioned at a greater distance. Yet, this arrangement is similar to the structure observed in Sordariomycetes. This remarkable finding challenges the traditional view of conserved flanking sequences within mating-type loci and their evolutionary role in *MAT* genes, diverging from the structure observed in the model organism *N. crassa*. This phenomenon calls for reevaluation of the significance of conserved flanking sequences in the evolution of *MAT* genes within the broader framework of mating-type loci.

## Figures and Tables

**Figure 1 jof-09-01202-f001:**
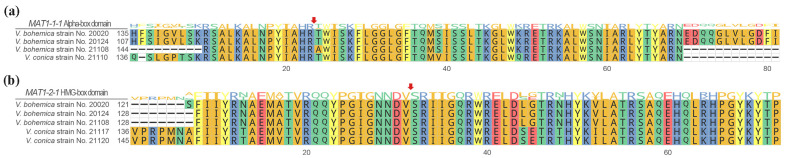
Alignment of part of the (**a**) *MAT1-1-1* MATalpha_HMGbox domain and (**b**) *MAT1-2-1* HMG box domain. The consensus sequences were based on a 60% similarity cutoff, while the sequence logo illustrates the relative representation of each amino acid per position. Red arrows indicate the positions of conserved introns. The figure was produced using the R package ggmsa; details of the sequences are available in the Appendix A.

**Figure 2 jof-09-01202-f002:**
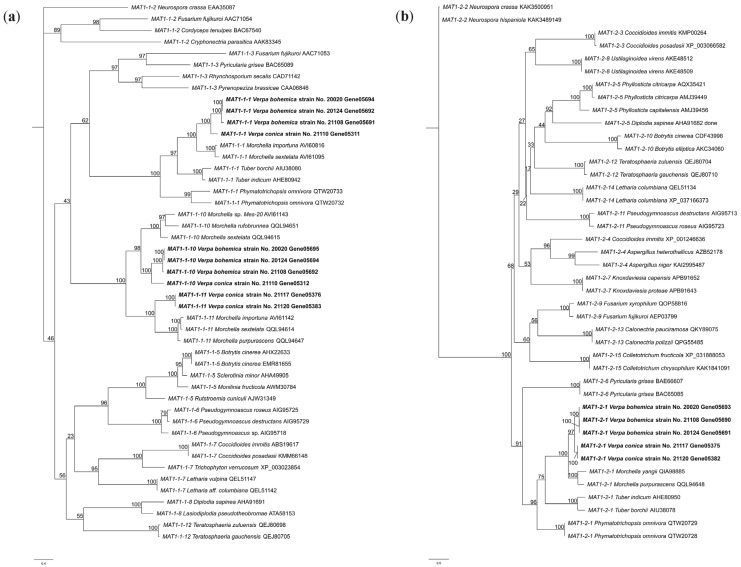
Phylogenetic trees of (**a**) *MAT1-1* and (**b**) *MAT1-2* genes constructed using IQ-TREE [30].

**Figure 3 jof-09-01202-f003:**
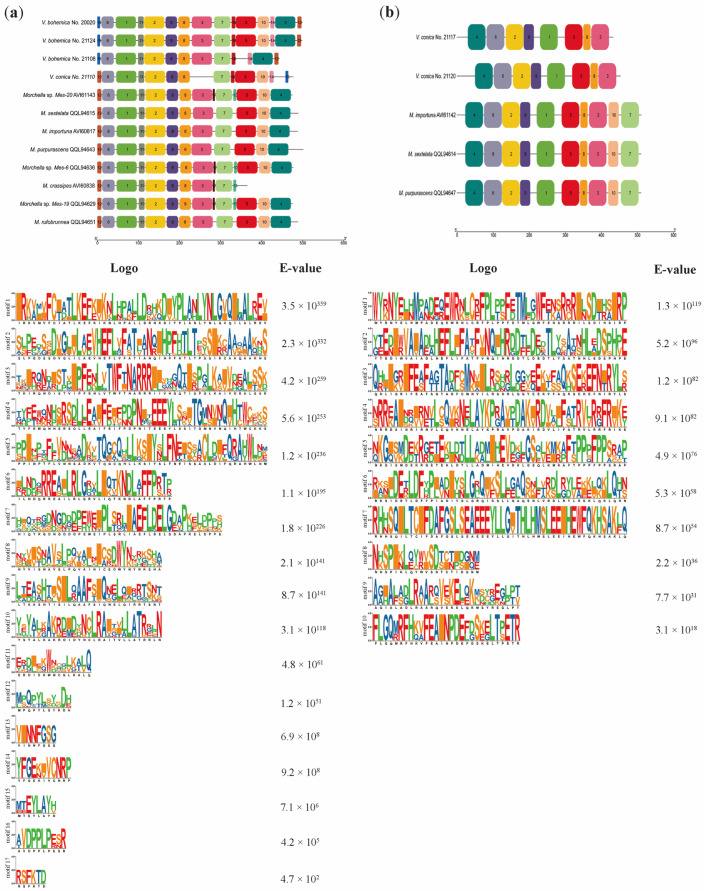
Distribution of conserved motifs of (**a**) *MAT1-1-10* and (**b**) *MAT1-1-11* proteins from *Verpa* spp. and *Morchella* spp. The scale bar at the bottom indicates the protein lengths, and sequence logos for each conserved motif are displayed below. Appendix A provide detailed sequences for each motif.

**Figure 4 jof-09-01202-f004:**
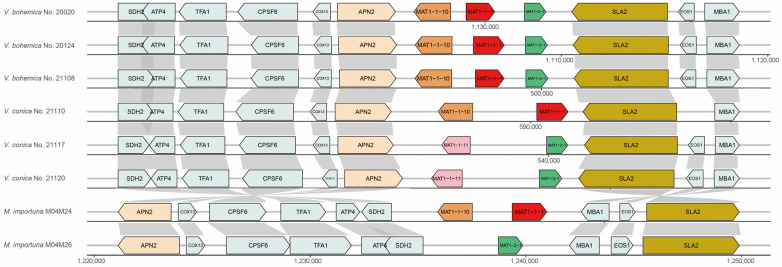
Genome maps of *MAT* loci for six *Verpa* species and two *Morchella* species. The names of the genes are labeled on the map and key genes are labeled with special colors. Genome maps of *MAT* loci were created using R package gggenes using fixed-length parameters.

**Figure 5 jof-09-01202-f005:**
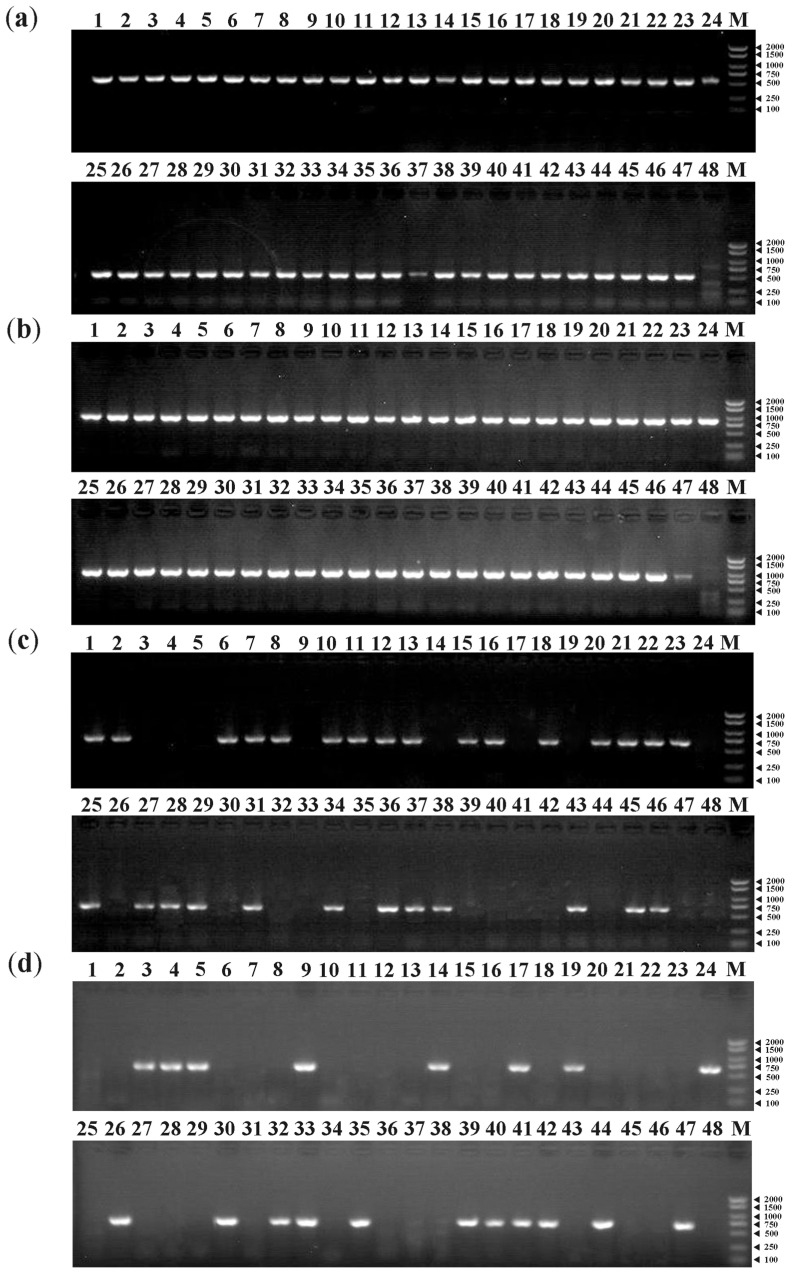
*MAT* gene PCR products of single-ascospore populations No. 20020 and No. 21120: (**a**) *MAT1-1-1* gene PCR products of No. 20020-1 to No. 20020-47; (**b**) *MAT1-2-1* gene PCR products of No. 20020-1 to No. 20020-47; (**c**) *MAT1-1-1* gene PCR products of No. 21120-1 to No. 21120-47; and (**d**) *MAT1-2-1* gene PCR products of No. 21120-1 to No. 21120-47. The first lane represents the parental strain and the 48th lane contains ddH_2_O, serving as a negative control in all gel images. ‘M’ denotes DNA marker 2000 (2000, 1500, 1000, 750, 500, 250, and 100 bp).

**Table 1 jof-09-01202-t001:** Ascocarps used in this study.

Ascocarp	Species	Origin	Genbank
No. 20020	*Verpa bohemica*	Shennongjia Hubei	JAUFSG000000000
No. 20124	*Verpa bohemica*	Linxia Gansu	JAUFSF000000000
No. 21108	*Verpa bohemica*	Arba Sichuan	JAUFSE000000000
No. 21110	*Verpa conica*	Pengshui Chongqing	JAUFSD000000000
No. 21117	*Verpa conica*	Tiaiyuan Shanxi	JAUFSC000000000
No. 21120	*Verpa conica*	Xinyuan Xinjiang	JAUFSB000000000

**Table 2 jof-09-01202-t002:** Primer sequences and PCR cycling parameters used in this study.

Gene	Code	Sequence (5′–3′)	Annealing Temperature (°C)	Product Size (bp)
*MAT1-1-1* gene of *V. bohemica*	b1f	TCTCGCAGAAAGCCACGATT	61	481
	b1r	ATCCTCATTGCGAGCGTAGG		
*MAT1-2-1* gene of *V. bohemica*	b2f	CTCCATCAAACGTGCGGTTC	61	654
	b2r	GGGGGTTTCTTGTCCGAAGT		
*MAT1-1-10* gene of *V. bohemica*	b10f	GGGCCTATCAATCAACGCTT	60	396
	b10r	AGGTAGCGTCTCAATGGGGT		
*MAT1-1-1* gene of *V. conica*	c1f	ATCTCGCAGAAGTCCACGATT	61	580
	c1r	TTGATACCACAGCCGAGAAGG		
*MAT1-2-1* gene of *V. conica*	c2f	GTCGTGACTAAAACCGGCCT	61	500
	c2r	TCATTGCTGCGGTGGATCTT		
*MAT1-1-10* gene of *V. conica*	c10f	CGCGTCGTTAGTAGTGCGAT	60	819
	c10r	GAAGCTCAGTCCGGTGACAT		
*MAT1-1-11* gene of *V. conica*	c11f	TGGACGAAACGCTGGACTTT	61	648
	c11r	CCTTCTGGAACCCTGCAAGT		

## Data Availability

Data are contained within the article and Appendix A.

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
