# Peer review of "Structure of the Mating-Type Genes and Mating Systems of Verpa bohemica and Verpa conica (Ascomycota, Pezizomycotina)"

_jof, 2023, doi:10.3390/jof9121202_

Round 1

Reviewer 1 Report (Previous Reviewer 1)

Comments and Suggestions for Authors

I appreciate the efforts of the authors to improve the quality of the manuscript, and can see a general improvement. However, there are several pervasive issues that still plague this study and I mention the two biggest ones here.

First is an issue with the gene models, and therefore the core data, used in the study. The description of the gene data is unclear and in the current form impossible to be true. What the authors present in the text and tabulate in the supplementary information do not add up (I have left a detailed explanation in the attached PDF document). If the gene models were as expressed in the supplementary table, then none of the MAT proteins would be possible. And if the gene models are not correct, it casts serious doubt on the rest of the findings.

Secondly, I have several issues relating to the inclusion of the phylogenetic analyses. These concerns are laid out in the document as well. Because the phylogenetic analyses take center stage in the results and discussion, it forms a large part of the manuscript. However, these are at best not applicable to the stated aim, and at worst completely incorrect.

I hope that the comments made will assist the authors in updating and improving the study.

Comments on the Quality of English Language

I am sorry to say that the English presented is not of a high quality. There are some poor choice of word, as well as syntax and grammatical errors. This also contributes to a loss in the quality of the work. I have highlighted just some examples in the abstract, but note that this is pervasive throughout the paper. I highly recommend that this should be addressed.

Author Response

Review Response Letter

Replies to the reviewers’ comments

  We would like to thank the reviewer for careful and thorough reading of this manuscript and for the thoughtful comments and constructive suggestions, which help to improve the quality of this manuscript. We sincerely hope that the revised manuscript will meet your approval. In this response letter, our responses have been shown in blue font.

Reviewer #1: I am not sure what is meant by "their fundamental study". Please clarify.

Author reply: Thanks for the suggestions. We have added detailed descriptions, and the original sentence in the manuscript has been changed to “However, fundamental research on their mating systems, the key aspects of their life cycle, re-mains scarce.”

Reviewer #1: What does refer to?  Please provide specific details of the procedure followed, or some reference to this technique in literature or in the current manuscript.

Author reply: Thanks. The original manuscript intended to convey that after gradient dilution, the liquid was transferred to new PDA plates. The original manuscript has been revised to “After the process of gradient dilution [22], three consecutive low-concentration gradient spore suspensions were independently applied onto three separate PDA plates and subjected to the same incubation conditions mentioned earlier.”

The reference was : [22]  Brock T.D., Madigan M.T., Martinko J.M. Brock biology of microorganisms. Benjamin Cummings,San Francisco, 13th edn, 2012.

Reviewer #1: Is this the file size (i.e. 4 gigabytes)? Or is it meant to be the number of bases produced (i.e. 4 gigabases). It would be better to report the latter. Is this the file size (i.e. 5 gigabytes)? Or is it meant to be the number of bases produced (i.e. 5 gigabases). It would be better to report the latter.

Author reply: Thank you for your query. The '4G' mentioned in the manuscript refers to the total number of bases generated, which is 4 gigabases (Gb). We apologize for any confusion caused and we have used 'gigabases (Gb)' to clarify the expression in the text.

Reviewer #1: Because the annotated genome is used in downstream analyses, and was also submitted as a genome sequence to the public repository, it would be prudent to provide details on how the annotation was done. How was the 4 gigabase worth of RNA data used for predictions. How was the ad initio annotations done? Surely this was not all done manually?

Author reply: Thank you for your insightful suggestions. We have addressed the issue by making corrections and providing a detailed explanation of the process of gene prediction and annotation. The content is as follows:

Gene prediction was performed using geta v2.6.1 (https://github.com/chenlianfu/geta), combined with ab initio, homologous protein alignment and RNA sequencing transcription alignment, followed by Nr annotation using a local Nr database, and custom scripts were utilized to organize and summarize the results into the final gene annotation file.

Reviewer #1: For this revision, I do not have access to the supplementary material. However, please ensure that somewhere sufficient information is supplied to allow a reader to identify and obtain a full genome sequence only using this manuscript.

Author reply: Thanks. After correction, we have provided the accession numbers in the text. The content is as follows:

The sequences of the two Morchella species [19] used for comparison were obtained from public genomic databases at NCBI with accession numbers QORM00000000 and QOKS00000000, respectively.

Reviewer #1: Please be specific here. What exactly were you using for the phylogenetic analysis? Was it all the mating-type genes, or just some of them? Was it the genes, or the proteins? What did you include as ingroups and outgroups? In addition, a serious flaw is the lack of clarity on the function of the phylogenetic analysis to the wider study. Why do these analyses in the way it was done?

Author reply: Thank you for your insightful suggestions. The primary objective of constructing a phylogenetic tree in this study was to validate the accuracy of the nomenclature for MAT genes in Verpa. In the constructed phylogenetic tree, different MAT proteins exhibited divergent branching patterns, characterized by the clustering of identical MAT proteins together. The purpose of this tree was not to demonstrate evolutionary relationships but to serve as corroborative evidence for the correctness of MAT gene nomenclature. We constructed a new maximum likelihood phylogenetic tree without setting an outgroup. To enhance clarity, we have performed a thorough phylogenetic analysis using the protein sequences of MAT1-1 and MAT1-2 in the latest version. Consequently, we have made the following modifications to the original manuscript and updated the corresponding graphics.

“As shown in Fig. 2, identical MAT proteins were clustered together. The analysis revealed that the MAT1-1-1 genes from the genus Verpa formed a cluster with 100% support, subsequently clustering with those of the genus Morchella, both belonging to Morchellaceae, and collectively with the genus Tuber (Tuberaceae) and Phymatotrichum (Rhizinaceae) in the order Pezizales. The corresponding MAT1-1-10 and MAT1-1-11 genes of the genus Verpa also clustered with those of the genus Morchella, with both genes clustered under independent branches under 98% and 100% support, respectively (Figure 2a). Moreover, the MAT1-2-1 gene from Verpa exhibited a high-support cluster with its counterpart in Morchella, Tuber and Phymatotrichum (Figure 2b). This evidence substantiates the accuracy of the designated nomenclature for MAT genes within Verpa spp..”

Reviewer #1: What was concatenated? It seems from the results if the trees consisted of datasets (nucleotide/proteins) of each gene aligned against each other? No concatenation was used?

Author reply: Thanks for the suggestions. After review, it was discovered that there were errors in this section of the original manuscript, which have now been corrected. The content is as follows:

To confirm the accuracy of the MAT proteins in Verpa, published MAT protein sequences were selected to construct a phylogenetic tree [17-18]. All sequences underwent alignment using MAFFT version 7.508 [29] with options --maxiterate 1000 --retree 1 -localpair. Maximum likelihood (ML) phylogenies were generated from the alignment in IQ-TREE v2.2.0 [30]. Branch supports were assessed using the ultra-fast bootstrapping method with 1000 replicates [31].

Reviewer #1: I am sorry to mention this again, but this section is still confusing. It refers to two introns in the gene, and then reports the length of the introns (according to what is written). However, this does not make sense. A check of the supplementary material adds some context. It appears that the lengths reported are gene lengths not intron lengths, so that is wrong here. But the gene models then also do not work. As example, isolate 20020 in Supp table 1 is reported as 1393 bp in length with 2 introns of 66 and 56 bp. If you do the math, that means the coding region will be 1393-66-56=1271bp in length. A codon has 3 bases, so the resulting protein should be 1271/3=423.67 aa in length. Yet it is given as 329 aa?

The same appeears true for all other MAT1-1-1 genes, as well as all other MAT genes. To make my point even stronger, compare MAT1-1-1 reported from 20020 and 21108. The 21108 gene is shorter and has a longer intron 2 (83 vs 86 bp), but yet produces a protein that is longer (348 aa) that that of 20020 (329 aa). I am not sure if the issue lies with the data capture into the table, or if this is a more substantial issues related to the annotations.

Author reply: Thank you for your careful review. You are right. We used incorrect descriptions and incorrect data in the original manuscript. The new revisions and supplementary materials are as follows.

“The gene structure and length vary inter-species and intra-species of Verpa. The predicted MAT1-1-1 genes in V. bohemica strains No. 20020, No. 20124, and No. 21108, as well as in V. conica strain No. 21110, each exhibited the presence of two introns, with the intron length as 57-66 and 56-57 bp, respectively. The lengths of these genes were different, measuring 1112 bp, 1112 bp, 1163 bp, and 1182 bp, respectively. The predicted MAT1-1-1 genes in Verpa spp. encoded proteins with the following amino acid sequences: 329 aa (No. 20020), 329 aa (No. 20124), 346 aa (No. 21108), and 355 aa (No. 21110) (Supplementary Table S1). The inferred amino acid sequences of these four Verpa strains all possessed a conserved MATalpha_HMGbox domain (GenBank: cl07856) with the E-values ranging from 9.15e-05 to 2.7e-05 (Figure 1a). In the case of V. bohemica, all three predicted MATalpha_HMGbox domain spanned by the first intron with the length of 66 bp, whereas it was interrupted by a 57 bp intron in V. conica No. 21110.

Similarly, the predicted MAT1-2-1 genes in V. bohemica strains No. 20020, No. 20124, and No. 21108, as well as in V. conica strains No. 21117 and No. 21120, each exhibited the presence of three introns. These genes exhibited varying lengths—namely, 1073 bp, 1073 bp, 1073 bp, 1081 bp, and 1081 bp—and their coding regions translated into protein sequences of 281 aa (No. 20020), 281 aa (No. 20124), 281 aa (No. 21108), 305 aa (No. 21117), and 305 aa (No. 21120), respectively (Supplementary Table S1).”

“All three strains of V. bohemica possessed MAT1-1-10 genes with five introns, each with a length of 1782 bp. However, the MAT1-1-10 gene of V. conica strain No. 21110 contained three introns with a combined length of 1686 bp. The MAT1-1-10 genes in Verpa spp. encoded proteins with the amino acid length as 498 aa (No. 20020), 498 aa (No. 20124), 441 aa (No. 21108), and 507 aa (No. 21110). These hypothetical MAT1-1-10 proteins from three V. bohemica strains showcased amino acid identities surpassing 35% with the Morchella genus, while those of V. conica strain No. 21110 had amino acid identities of 38.60% (e-value = 1e-114) and 37.94% (e-value = 2e-104) with Morchella sp. Mes-20 (AVI61143) and M. rufobrunnea (QQL94651), respectively (Supplementary Figure S3). In contrast, the MAT1-1-11 genes found in V. conica possessed three introns measuring 1619 bp and 1995 bp in length, with the coding region encoding proteins of 445 and 424 amino acids, respectively (Supplementary Table S1).”

Reviewer #1: I am not sure what is meant with "specificity". Motif 17 appears common to both Morchella and Verpa species, so it is not specific.

Author reply: Many thanks for your careful review. There were errors in the original manuscript. Motif 17 is unique to Morchella, not Verpa. The motif unique to Verpa is motif 16. The issue has been corrected in the revised manuscript.

Reviewer #1: I have highlighted above where Motif 3 is claimed to be exclusive to V. bohemica, but now it is also in multiple Morchella species. Please address this inconsistency.

Author reply: Thank you for your insightful suggestions. The reviewer is right. The issue has been addressed in the revised manuscript by removing the sentence in question.

Reviewer #1: Why these three motifs? All the motifs apart from 7 and 10 are common to the proteins included, and many of these have higher statistical support.

Author reply: Thank you for your query. In considering motif1, motif2, and motif9 as domains, the decision was primarily based on the continuity of the sequences and the higher e-value of motifs in these regions. Due to its considerable variability, motif11 was not taken into account.

Reviewer #1: Please label the figure as was done for the gel images included in the manuscript (lane numbers)

Author reply: Thanks. The issue has been corrected in the revised manuscript.

Reviewer #1: I am not sure what the relevance is of including a description of the N. crassa MAT locus? This species is not related to the species discussed here, and do not have a comparable MAT gene complement.

Author reply: Many thanks for your careful review of our manuscript. The reviewer is right and we have decided to remove this section.

Reviewer #1: This sentence is not clear. Are the authors arguing that the phylogenetic analyses presented here can be used to confirms species boundaries, or were they designed to confirm the names assigned to the MAT genes? Either way, the analyses are not sufficient to answers any of these questions.

Author reply: Thank you for your insightful suggestions. Belfiori et al. stated, "Fast-evolving genes in the MAT locus are thus good candidates for tackling taxonomic and phylogenetic questions and for better delineating species boundaries within the Tuber genus." After careful consideration, we believe this statement may be somewhat controversial, and have therefore removed it. The revised content is as follows.

In this study, phylogenetic trees were constructed using different MAT1-1 and MAT1-2 protein sequences to corroborate the accuracy of the naming of MAT genes in the genus Verpa.

Many thanks again for the your careful, thorough, and professional review of our submission, which further improves the quality of the paper.

Thank you again.

Sincerely,

Wenhua Sun, Liu Wei, Yingli Cai, Xiaofei Shi, Liyuan Wu, Jin Zhang, Lingfang Er, Qiuchen Huang, Qi Yin, Zhiqiang Zhao, Peixin He* and Fuqiang Yu*

Reviewer 2 Report (Previous Reviewer 2)

Comments and Suggestions for Authors

As this is a resubmission, the authors have made a number of improvements to their manuscript.  I believe that they have addressed the major concerns that i had with the work in sufficient detail.  I do no object to proceeding with this work by the production editor for publication at this time.

Comments on the Quality of English Language

The language is appropriate 

Author Response

Review Response Letter

Replies to the reviewers’ comments

I would like to express my sincere gratitude for your valuable feedback and suggestions on our submission. Your insights were instrumental in guiding the revisions that greatly improved the quality of the work. I am thrilled to hear that the concerns you had have been effectively addressed in the resubmitted manuscript. Your expertise and thoughtful guidance have been immensely valued throughout this entire process.

Thank you once again for your constructive critique and unwavering support.

Sincerely,

Wenhua Sun, Liu Wei, Yingli Cai, Xiaofei Shi, Liyuan Wu, Jin Zhang, Lingfang Er, Qiuchen Huang, Qi Yin, Zhiqiang Zhao, Peixin He* and Fuqiang Yu*

Reviewer 3 Report (Previous Reviewer 3)

Comments and Suggestions for Authors

I have a concern with the claims about protein motifs 'revealed' in the study.  When I think of such motifs that may have relevance to specific enzymatic or perhaps DNA binding such as HTH, signals sequences , sites for specific modifications etc.  There is nothing I could find concerning the relevance of the motifs they claim to have identified.  

Comments on the Quality of English Language

see the returned document for suggestions 

Author Response

Review Response Letter

Replies to the reviewers’ comments

  We would like to thank the reviewer for careful and thorough reading of this manuscript and for the thoughtful comments and constructive suggestions, which help to improve the quality of this manuscript. We sincerely hope that the revised manuscript will meet your approval. In this response letter, our responses have been shown in blue font.

Reviewer #3: How are the motifs defined, ie are they just conserved amino acids at a specific position or are do they motifs known to have a function? Relevance?

Author reply: Thanks. The motifs in this study are defined based on conserved amino acids motif at a specific position in a specific population, and the specific function of these motifs is currently unknown.

Reviewer #1: isn’ this enough to conclude it is homothallic?

Author reply: Thank you for your query. In this study, we primarily identified the MAT locus from Verpa genomic data and discovered that V. bohemica possesses both MAT1-1-1 and MAT1-2-1, whereas V. conica only has one of either MAT1-1-1 or MAT1-2-1. Thus, it's demonstrated that V. bohemica is homothallic, while V. conica is heterothallic in genomics. Then, PCR amplification of MAT genes was conducted on the monosporal population to provide additional confirmation of their life-cycle type. The obtained results align with the findings from the previous genomic bioinformatics analysis. Therefore, V. bohemica is homothallic, while V. conica is heterothallic.

Reviewer #1: Doesn’t this match the clear “homothallic” observation for ascospore production from a single ‘spore’ as stated in the beginning?

Author reply: Yes, you are right. Many thanks for your careful review of our manuscript.

Many thanks again for your meticulous review, revision of grammar details, and the highly professional feedback on our manuscript. We sincerely appreciate the time and effort you dedicated to offering valuable insights and suggestions.

Sincerely,

Wenhua Sun, Liu Wei, Yingli Cai, Xiaofei Shi, Liyuan Wu, Jin Zhang, Lingfang Er, Qiuchen Huang, Qi Yin, Zhiqiang Zhao, Peixin He* and Fuqiang Yu*

This manuscript is a resubmission of an earlier submission. The following is a list of the peer review reports and author responses from that submission.

Round 1

Reviewer 1 Report

Comments and Suggestions for Authors In the study by Sun et al, the authors present their findings on the  mating system of two Verpa species using full genome sequencing  complimented by additional molecular analyses. The authors identified the MAT1-1-1, MAT1-2-1, MAT1-1-10 and MAT1-1-11 genes in the mating type locus, and described these species as homothallic (V. bohemica) or heterothallic (V. conica). Overall, the results seem interesting, and should be if interest to anyone studying this group of fungi (the morals) or those with an interest in the mating-type genes or evolution of mating systems. However, a lack of thorough analysis makes it difficult to gain confidence in the methods used for the study, the results presented and the conclusions drawn from these findings. Although the overall results appear valid, more rigorous analysis is needed to support these. In addition I believe that the manuscript will also benefit extensively for better presentation of the findings. I offer the following critique on the different sections: 1. The introduction consists of only two paragraphs, and are not sufficient to introduce the study. I would like to see more background information on the Verpa species. This can include, but may not be limited to, economic importance, ecological importance, phylogenetic placement. Similarly, a large part of the subsequent paper is spent on comparisons with Morchella and Tuber, yet very little information on the relationship to these species are presented. Additionally, there should definitely be more information provided on the MAT gene work that has been done in these species. 2. Overall, the presentation and detail of the materials and methods could be improved. I rpovide some examples: a. Field samples were collected and "All of them were subjected to tissue isolation to obtain pure cultures of the strains. What does this mean? And how was this done? b. DNA is extracted using CTAB, phenol-chloroform and RNAseA. But how is the DNA recovered? I see no ethanol step, or the use of a kit to obtain pure DNA samples. c. One is left to deduce that both DNA and RNA is sequenced. All that is stated is that RNAse treatment "resulting in pure DNA. Total RNA was extracted by TRIzol® Reagent (Invitrogen) and purified with Plant RNA 79 Purification Reagent (Invitrogen)". Surely this cannot be it? d. Where was the genome sequencing done? e. I struggle to follow the flow of information. The cultures described in a. above was used for sequencing, but then section 2.2 describes the generation of monosporic isolates from fruiting bodies that has now stated relation to these six isolates. And then in the same section mention is made of submitting sequencing data to SRA at NCBI. And only then is the genome assembly and analysis described. How does all of this relate to each other? f. I see no details on QC control for any sequencing. g. Provide accession information for the query sequences from Morchella and Tuber used. f. What is a whole-genome scan - lines 111-112. h. Genes cannot be used as queries in blastp searches. i. The primers referred to in section 2.4 - what were their functions? Obviously I can work it out from Table 2, but a sentence to say why this was done and what you hope to achieve would be good. j. Sections 2.5 and 2.6 are inadequate. I gather the Genome comparisons were actually gene comparisons, which are not very useful. The phylogenetic analysis is never presented in the results, and there is no information on the genes used for the phylogenetics. 3. The results section has many parts that fits more with discussion, and it turns out there are results in the discussion section. This should really be corrected. a. You used the position of the SLA and APN genes to find the MAT locus? So why the download the MAT genes from the related species. Does this have to do with the "whole-genome scan"? b. In lines 152 and 153 it is stated the "MAT1-1-1-1 gene, characterized by a pair of introns measuring 1393 bp, 1416 bp, and 1101 bp". I am not even sure I know what this means. How are three things a pair? And how does one describe a gene in terms of introns? This MAT1-1-1 description, together with the frequent confusing reference to Introns when other MAT genes are described, makes me consider that the authors are maybe misunderstanding the use of the word "intron", and that this should rather be a reference to "exon", potentially. c. I do not see how the similarity values in lines 157-161 were derived. The figure referred to shows a dot plot, and does not give any useful information. Also, the only dot plots referred to in the Materials and Methods are a genome comparison. However, there is no genome comparison done. d. I am wondering about the MAT1-2-1 gene. There are several HMG box genes often present in the fungal genome. Although the Phylogenetics analysis in Figure 4b refers to MAT1-2-1, I am worried about the outgroup there. I would suggest that the authors identify if an intron is present in the HMG box across a conserved serine codon (I propose it will be across the serine amino acid - residue 10 in figure 1b). This has been identified in a number of fungi. e. Section 3.2 is difficult to follow. Firstly, please describe the results of the BLAST searches for V. bohemica in terms of e-values as well. Also, make it clear if the reported matches were the top results - the question here is what are these genes, and part of the answer is what does it match most closely in the Genpept database. Did they match anything else better? I do however concur that these would likely match known MAT genes previously described from Morchella. Also, do not refer to "new open reading frames". These should be additional reading frames, and then describe the process of identifying these. f. The MEME analysis used to identify "conserved regions" in the two secondary MAT genes are questionable to me. Firstly, I do not know how the analysis was done - it is not mentioned in the methods section. Secondly, the image is of low quality, so I am struggling to interpret it, and it is not included in the "high quality image versions" download. That being said, if only the MAT proteins from Morchella and Verpa was included, MEME analysis would, with a high confidence (e-value) predict motifs in regions with high sequence similarity. These genes likely were derived by common descent in species that are closely related. As such, I offer the different opinion that the MEME motifs represent sequences that are common due to descent, and not common due to functional restrained, as one would expect from a conserved domain. This should be addressed. g. Lines 241-249: The argument that lower or higher levels of amino acid sequence similarity directly relates to evolutionary speed seems a bit simplistic. Different genes have different rates of evolution, and to then infer idiomorph evolutionary rate on the evidence provided should be better justified. h. In figure 3, there are several bands in the negative controls (no 48) for gels A and B. Also, no negative control was included in gels C and D, which becomes more significant when there are bands in lanes 19 and 23 of Gel C that then questions the results compared to image D. All of this should be corrected or explained. i. Similarly, the bands in supplementary figure 6c lanes 5, 6, 10 and 39 casts doubt on the results presented. Also, there does not seem to be a negative control included. j. Figure 4: Why is there such a discrepancy between the taxa for the two trees? Also, using MAT1-1-1 as an outgroup for a MAT1-2-1 phylogeny is strange, and in my opinion incorrect.   4. The discussion needs to be improved as well. a. The full first paragraph 283-302 should be rewritten based on the comment about evolutionary rate and the phylogenetic analysis I made earlier. b. The discussion in section 4.2 should be improved. Specifically, the use of idiomorphs are incorrect and does not reflect the results. Secondly, the authors should contextualize the results better, especially supporting the presence of MAT1-1-11 (a MAT1-1 gene) in the MAT1-2 idiomorph of V. conica. I am aware that there are such exceptions in some Morchella species (the study of Chai et al), and therefore crafting a clearer discussion should be possible. c. Lines 354-374 is only background information and does not relay any information on the findings in this study. This should be addressed. d. The extensive discussion on the flanking genes in the discussion is not well supported by any results.   Other comments: 1. Lines 47-48: The two genes are not called idiomorphs. The mating-type "alleles" of the MAT locus in heterothallic species are called idiomorphs. An idiomorph can have one gene or multiple genes. 2. Lines 54-55: MAT1-1 genes are up to 12, and MAT1-2 genes are up to 15. Please see: Wilson AM, Wilken PM, Wingfield MJ, Wingfield BD. Genetic Networks That Govern Sexual Reproduction in the Pezizomycotina. Microbiol Mol Biol Rev. 2021;85(4):e0002021. doi:10.1128/MMBR.00020-21 3. Overall, more attention should be paid to trace the cultures throughout the study. For example, in lines 152-155 the statistics are given for the MAT1-1-1 genes in three strains, but it is not clear which size fits with the gene from which strain. The reader needs to access a supplementary table for this information. 4. Assuming the alignment referred to in lines 157-161 was done on nucleotide level, I would recommend that this be reconsidered. MAT genes are notoriously diverse at the nucleotide sequence level, and often at the amino acid alignments as well. This can be clearly seen in Supplementary figures 2, 3, 4 and 5. In general, a BLASTp analysis and phylogenetic comparison may be more informative. 5. Remove "Bulleted lists look like this:" from line 180. 6. Overall, the authors should pay attention to their use of MAT terminology throughout. The term idiomorph is frequently used incorrectly, and is not applicable to homothallic species. Also, genetic usages of the MAT gene names should be italics (locus, genes and idiomorphs) while proteins should not be italicised. Comments on the Quality of English Language Although the use of English is acceptable, I would recommend that the manuscript could benefit from some English editorial work. The use of some words are not suitable to the context, and minor to moderate edits could improve clarity.

Reviewer 2 Report

Comments and Suggestions for Authors

The reviewed manuscript, titled “Structural Evolution of the Mating Type Genes and Mating System Analysis of Verpa bohemica and Verpa conica (Ascomycota, Pezizomycotina)” by Sun et al. is a largely descriptive work that is focused on identification of the mating locus in Verpa bohemica and Verpa conica.  This work has a number of elements that are interesting and could potentially warrant publication at some point, however overall, this work is largely descriptive, poorly structured and written, and provides little insight.  The title is misleading, as this work provides a rather basic structural comparison and a phylogenetic tree, but with little insight into ‘structural evolution’. 

Major issues

Introduction: much too brief and truncated.  This reviewer would like to encourage the authors to clearly – and thoroughly – explain the mating system and background for this work.  As written, the paragraphs are so brief that they jump topic rather quickly from sentence to sentence!  For example, the second paragraph of the introduction, which begins on Line 57, the authors jump from a rather basic description of the genotype of heterothallic and homothallic ascomycetes, to the rate of evolution of sex determining genes, to the Morchellaceae family of fungi.  As written, this jumps around and does not lay a solid foundation for the background and study performed.  This reviewer strongly encourages the authors to include rationale, context, and importance to this section.  Split it up as needed to provide a clear – and not simply concise – introduction and background section for the readers.

Line 150: “the mating-type gene was consistently found only around the APN2 and SLA2” – italicize gene names as is standard convention.  Based on this, are the authors stating that this is the locus with the actively expressed MAT gene(s)?  The authors discuss the ascomycetes mating system, which can involve switching of the actively expressed loci (based on HO activity and the copy and paste type mechanism from the hidden mating loci which are the repressed site of these genes, as in S. cerevisiae).  I am not as familiar with this species – as the authors state earlier that this is a less studied organism – and would appreciate context.  As written, I am unclear about the significance of this (if any).

Line 152 – this paragraph and the next are simple descriptions that could be presented in a clearer manner.  Perhaps a table, or schematic of the loci and gene structure?  Reading %s and positions of introns is not the best – or most appropriate – way to convey this information.

Line 181 – Figure 1.  Have the authors made a comparison to the known identity and AA structure of the two domains in the picture from previously characterized motifs?  This figure appears to be an alignment and motif from these alignments, with the consensus motif derived from these.  Are the actual motifs identical in size, at 50 AA each? Are there AA that are known to be more conserved than others, and are the deviations (or conservations) consistent with the literature?  This section is largely descriptive, but presented as a list of facts, rather than an analysis.

Line 250 – This figure is organized and formatted in a manner that does not provide information to the reader.  The font is consistently too low, the conserved regions are illegible, and there is no way to link the color of the section to the logo plots that are presented below each.  The logo plots are illegible, and many seem to have a large orange region that is blocked off that does not appear to be a letter.  I honestly do not know what to make of this, or how this provides any evidence in support of the author’s claims as provided for peer review.  A rework of this figure, and not simply a re-formatting, would be needed to make this clear and useful.

Line 275 – Figure 3.  The authors are correct in their labelling and description of what was done – however I am not sure that this adds anything to the paper that hasn’t already been covered in a better manner earlier (e.g. the sequencing data).  I feel that this is completely unnecessary and adds nothing. 

Line 303 and 350 – Figures 4 & 5 – these would be better served as a component of the results section prior to discussing them in this section.  The gggplots figure, while accurate, could be formatted better to make this sized and labelled for clarity. 

Minor details

Line 28: “encompasses both MAT1-1-1 and MAT1-2-1 gene” – genes are typically italicized for clarity.  A lack of italicization refers to a protein.  Please review the manuscript and revise according to standard conventions.

Line 33: “gene in the idiomorph, while the other” – there is a formatting issue here.  It may be the result of conversion to PowerPoint, but there is a font and size difference in the text.  Please double check this section during revision.

Line 31, 33, 48: Check the sentence construction around the word idiomorph. 

“The term idiomorph is used to describe the situation in ascomycete fungi where alternative versions of the mating type locus on homologous chromosomes are completely dissimilar and encode unrelated proteins (Metzenberg and Glass, 1990)”

It appears to this reviewer that the authors may have been a little imprecise in the use of this word.  Please review the sentence construction around the usage for clarity and proper grammar.  For example:

Line 31 - “strains of Verpa conica, No. 21110 strain harbors a MAT1-1-1 gene, while the other two strains bear the MAT1-2-1 gene in their MAT idiomorph” – this usage is improper – the authors are comparing strain differences rather than the homologue differences.  This is one example, and the above lines have several additional imprecise usage.  Please review and revise accordingly throughout.

Line 64: “fungi within Morchellaceae” – Morchellaceae is a family name, I believe that this should be italicized.

Line 72: Please revise the materials and methods for completion and clarity to allow reproducibility of the work done.  As written there are a substantial number of omissions in the methodologies.  Furthermore, outline the construction of the libraries, as well as the accession information for the raw data (and processed data, if applicable) from an appropriate archive (sequence read archive, GEO, etc.).  This particular section is lacking significant amounts of information.  I found references to this later in the methods, please include these together for coherence.

Line 92: Even for standard media it is typical to provide details, e.g. PDA – what are the percentages of each component?  Was this purchased as a pre-mixed powder from a vendor? Etc.

Line 76: “and draft DNA was extracted using” – I am unclear what is meant by the authors when they are describing isolation of draft DNA.

Comments on the Quality of English Language

English has some fundamental grammatical and structural issues.  These can be resolved by the authors with a rigorous review of the document.

Reviewer 3 Report

Comments and Suggestions for Authors

There are suggested English edits on the returned MS.  Most very minor.  In Information on page 6 is confusing: it seems to say that introns are coding for several proteins but I assume you mean that the coding region(s) do that?

Also. I assume the protein sequences were tested versus Swiss-prot or similar databases; If not they should be and if so with no similarities that should be noted. Also it would be useful to provide information where it is known,  on the role the motifs play if found in other proteins.

Comments on the Quality of English Language

generally very good- suggested improvements are provided